# Cancer-Associated Fibroblasts: Mechanisms of Tumor Progression and Novel Therapeutic Targets

**DOI:** 10.3390/cancers14051231

**Published:** 2022-02-27

**Authors:** Ralf-Peter Czekay, Dong-Joo Cheon, Rohan Samarakoon, Stacie M. Kutz, Paul J. Higgins

**Affiliations:** 1Department of Regenerative & Cancer Cell Biology, Albany Medical College, Albany, NY 12208, USA; czekayr@amc.edu (R.-P.C.); cheond@amc.edu (D.-J.C.); samarar@amc.edu (R.S.); 2Biology & Health Science Program, Russell Sage College, Albany, NY 12208, USA; kutzs@sage.edu

**Keywords:** cancer-associated fibroblasts, tumor microenvironment, chemoresistance, SERPINE1, fibrotic stroma, TGF-β, cancer progression

## Abstract

**Simple Summary:**

The tumor microenvironment plays an important role in determining the biological behavior of several of the more aggressive malignancies. Among the various cell types evident in the tumor “field”, cancer-associated fibroblasts (CAFs) are a heterogenous collection of activated fibroblasts secreting a wide repertoire of factors that regulate tumor development and progression, inflammation, drug resistance, metastasis and recurrence. Insensitivity to chemotherapeutics and metastatic spread are the major contributors to cancer patient mortality. This review discusses the complex interactions between CAFs and the various populations of normal and neoplastic cells that interact within the dynamic confines of the tumor microenvironment with a focus on the involved pathways and genes.

**Abstract:**

Cancer-associated fibroblasts (CAFs) are a heterogenous population of stromal cells found in solid malignancies that coexist with the growing tumor mass and other immune/nonimmune cellular elements. In certain neoplasms (e.g., desmoplastic tumors), CAFs are the prominent mesenchymal cell type in the tumor microenvironment, where their presence and abundance signal a poor prognosis in multiple cancers. CAFs play a major role in the progression of various malignancies by remodeling the supporting stromal matrix into a dense, fibrotic structure while secreting factors that lead to the acquisition of cancer stem-like characteristics and promoting tumor cell survival, reduced sensitivity to chemotherapeutics, aggressive growth and metastasis. Tumors with high stromal fibrotic signatures are more likely to be associated with drug resistance and eventual relapse. Clarifying the molecular basis for such multidirectional crosstalk among the various normal and neoplastic cell types present in the tumor microenvironment may yield novel targets and new opportunities for therapeutic intervention. This review highlights the most recent concepts regarding the complexity of CAF biology including CAF heterogeneity, functionality in drug resistance, contribution to a progressively fibrotic tumor stroma, the involved signaling pathways and the participating genes.

## 1. Cancer-Associated Fibroblasts: Biology of the Fibrotic Tumor Stroma

Molecular and functional heterogeneity among fibroblast subpopulations contribute to the phenotypic complexity of the mesenchymal subsets evident in normal vs. inflamed tissues, in adaptive as compared to maladaptive wound repair and in the tumor microenvironment (TME) (e.g., [1,2,3]). The composition of the peritumor milieu varies depending on the specific malignancy but generally consists of a diverse complement of highly interactive resident and recruited cell types that coexist with the growing cancer in a hypoxic, progressively fibrotic, stromal matrix [4]. Cancer-associated fibroblasts (CAFs) constitute a significant fraction of the cellular repertoire of the TME engaging in a dynamic crosstalk with cancer cells, infiltrating tumor-associated macrophages (TAMs) and other stromal elements [5,6,7,8]. CAFs and TAMs are the most abundant nonmalignant cell types in the TME, particularly in aggressive desmoplastic tumors [9,10]. Cytokines produced by the malignant epithelium likely recruit resident fibroblasts to a CAF precursor state and, ultimately, to a “mature” CAF. Various CAF subtypes, in turn, support tumor progression and phenotypic transitions largely through paracrine signaling by secreted soluble factors including members of the transforming growth factor-β (TGF-β), insulin-like growth factor and interleukin families [7,11,12,13]. Such dialogue between CAFs and cancer stem cells promotes cellular plasticity and tumor progression, maintenance of cancer cell stemness, metabolic reprogramming, extracellular matrix (ECM) remodeling and metastasis [7,8]. Clarifying the critical pathways in the CAF–tumor interacting network may provide new venues for therapeutic intervention [14].

The term “CAF” comprises a group of several fibroblast subgroups, each influencing cancer progression somewhat differently depending on their spatial-mechanical properties, degree of senescence and expression of key tumor-modulating factors (e.g., TGF-β) [15,16]. Such heterogeneity, made more evident by data obtained by single-cell RNA_seq_ analysis (reviewed in [3]), contributes to the uncertainty as to the specific cell type or types that give rise to the CAF cohort or cohorts in various human malignancies. This likely reflects the overall difficulty in defining a “fibroblast” since there are no unique biomarkers that provide an unambiguous identification and strategies utilizing exclusion criteria may underestimate the diversity in fibroblast origins [17,18]. Several additional confounders complicate a precise molecular definition of CAF derivation. Among the most significant is the absence of specific lineage tracing-adaptable Cre drivers for normal fibroblasts or their CAF counterparts in mouse models of tumorigenesis and the realization that subsets of fibroblasts differ as a function of tissue localization and mobilization state [8,18,19].

There are considerable similarities between the development of pathologic fibrosis during injury repair and the emergence of a CAF-enriched tumor stroma, not the least of which involves the engagement of both the canonical (SMAD-dependent) and noncanonical (non-SMAD) arms of the TGF-β signaling network [3,15,20,21,22,23,24,25]. The growing appreciation for the congruities between fibrotic and neoplastic diseases suggests that targeting commonalities may have shared therapeutic utility [20]. Indeed, TGF-β1 mobilizes the Rho GTPase, mitogen-activated protein kinase, phosphoinositide-3-kinase and p53 pathways and upregulates the Hippo effectors YAP (*yes*-associated protein) and TAZ (transcriptional coactivator with PDZ-binding motif) (e.g., [20,26,27]). In vitro modeling confirmed that TAZ is necessary for TGF-β1-mediated fibrogenesis and vector-driven TAZ synthesis mimics aspects of the TGF-β1-induced phenotype including G_2_/M arrest and acquisition of a profibrotic program [27]. TAZ is required for maximal TGF-β1-mediated expression of the SMAD target gene encoding plasminogen activator inhibitor-1 (PAI-1), a potent profibrotic clade E member of the serine protease inhibitor family (SERPINE1) [27,28,29,30], and a similar involvement of YAP in TGF-β1-induced PAI-1 expression is evident in lung tumor cells [31]. KEGG analysis confirmed, moreover, that convergence of the TGF-β and Hippo signaling pathways regulates transcription of the fibrosis-inducing connective tissue growth factor (CTGF; CCN2) and SERPINE1 genes [32] (Figure 1). YAP knockdown effectively reduces levels of both CCN2 and PAI-1 (SERPINE1), while introduction of the constitutively active YAP^S127A^ construct increases PAI-1 expression in immortalized cell lines [33]. Although the underlying mechanisms remain to be determined, it is apparent that YAP/TAZ do not alter the rate of SMAD nuclear import or exit nor impact SMAD phosphorylation but may regulate SMAD nuclear levels by functioning, directly or indirectly, as retention factors and/or by altering TGF-βR activity. It is apparent, moreover, that YAP and TAZ integrate bidirectional responses between tumor and stromal cells functioning, thereby, as signaling hubs, perhaps as a consequence of increasing cellular tensile forces as well as the changing mechanical properties (i.e., progressive stiffening) of the TME [24,34,35].

## 2. CAF Functional Complexity

Single-cell RNA_seq_ studies identified five to seven fibroblast subtypes and several subpopulations just in the cutaneous compartment [37,38]. Functionally distinct disease-associated fibroblast subsets, moreover, can be found in a number of relatively common inflammatory disorders as well as in bleomycin-induced pulmonary fibrosis [39,40,41]. A similar diversity is evident in the tumor-associated fibroblastoid population (reviewed in [3]) with CAF plasticity, molecular variants and their individual pathophysiologic features contributing to tumor progression and chemoresistance [2,15,19,42,43]. CAFs secrete proinflammatory, growth- and immune-regulating cytokines, senescence-associated factors and ECM elements that collectively promote aggressive behavior, an attenuated response to therapy, and increased tumor cell survival and proliferation [2,3,8,13,44,45,46,47,48]. CAFs circumvent tumor chemo-/radiosensitivity by elevating DNA repair capacities and expression of antioxidants as well as multidrug resistance factors implicating CAFs as “abettors” of tumor progression [48,49]. CAFs in the renal TME, for example, express elevated levels of tryptophan 2,3-dioxygenase, which stimulates kynurenine secretion, leading to the upregulation of the aromatic hydrocarbon receptor and subsequent AKT/STAT3 pathway activation [50]. This directly impacts renal tumor progression as AKT inhibition attenuates tumor cell proliferation, while interference with STAT3 signaling blunts kynurenine-induced drug resistance and carcinoma cell migration.

Although several factors serve as fibroblast activators, members of the TGF-β and interleukin cytokine superfamilies are, perhaps, the predominant contributors to the creation of the highly fibrotic desmoplastic tumor stroma while also promoting cell migration and tissue invasion [20,23,51,52,53,54,55]. TGF-β signaling regulates both positively and negatively the core elements in the diverse repertoire of fibroblast functions [56]. TGF-β1 governs the myofibroblastic differentiation of vascular pericytes, resident fibroblasts and other TME cellular lineages while coordinating a program of pathologic ECM synthesis, increased tensional force and advancing fibrosis [20,57,58,59]. Secretion of TGF-β1 by CAFs also initiates a plastic response in neighboring cancer cells [13]. There is a clear inter-relationship among chronic inflammation, progressive tissue fibrosis and metastatic dissemination, particularly for malignancies of the head and neck, lung, pancreas, ovary, breast and colon with CAF-derived TGF-β as a major contributor to cancer stem cell self-renewal, tissue invasion and distal spread [54,60,61]. These particular tumors frequently exhibit a robust desmoplastic reaction typified by a marked accumulation of ECM with abundant numbers of embedded CAFs [62]. This dense fibrotic stroma increases matrix stiffness, which promotes an epithelial-to-mesenchymal transition (EMT) in cancer cells, more aggressive behavior and a poor prognosis [63,64,65,66,67,68]. Matrix stiffness, in fact, negatively correlates with the response to paclitaxel in several cell lines and in pancreatic ductal adenocarcinoma (PDAC), while increased environmental tension significantly decreases tumor sensitivity to gemcitabine [69,70,71].

CAFs mechanically contract the ECM; the resulting enhanced tension is communicated to neoplastic cells via integrin-mediated mechanotransduction [64,72,73]. CAFs utilize Rho-dependent signaling to reorganize the tumor matrix and create fibronectin-, collagen- and tenascin-C-rich guidance tracks to promote migration [74,75]. The leading cell in this motile cohort is usually a fibroblast that, unlike the follower tumor cells, employs Rho-ROCK signaling and protease-mediated stromal remodeling to effectively direct carcinoma invasion [74,76].

Spatial heterogeneity is also evident among CAF subtypes [77,78]. In PDAC, at least two different CAF phenotypes, defined based on expression of α-SMA and IL-6, are evident in specific regions in the tumor mass [79]. Depending on their location (i.e., the fibrotic and hypoxic center vs. peripheral areas), CAFs may be exposed to distinct environmental conditions (e.g., varying oxygen and nutrient levels, progressive changes in matrix tension) that will either alter or sustain a specific phenotype. A more matrix-secreting, high-α-SMA and low-cytokine (e.g., IL-6, IL-11)-expressing CAF population is enriched in the peripheral areas, while inflammatory-type CAFs (high IL-6 and IL-11, low α-SMA) are found in the fibrotic center [79]. CAF IL-6 expression is upregulated in response to IL-1 secreted by malignant cells, whereas production of TGF-β (by tumor cells) downregulates CAF surface IL-1R, favoring increased matrix secretion by this CAF subtype [80]. Targeting this loop may have clinical utility. Indeed, elevated levels of IL-6 in pancreatic stellate cells activate the JAK/STAT pathway and stimulate growth in PDAC [79]. Similarly, coculture of mouse pancreatic tumor organoids with IL-6-expressing pancreatic stellate cells significantly enhanced cancer cell growth, whereas IL-6-deficient stellate cells failed to support extended tumor survival [79]. Clarifying the impact of the dynamic CAF environment on tumor growth and metastasis may inform novel treatment strategies targeting specific CAF populations.

## 3. Heterotypic Spheroids: A Specialized CAF–Tumor Microenvironment

While solid tumors in most human malignancies disseminate via the vascular and lymphatic systems, ovarian and gastrointestinal neoplasms metastasize by means of a transcoelomic route [81]. In ovarian cancer (OVCA), EMT-induced loss of cellular cohesion facilitates a “shedding” event whereby certain tumor cells leave the primary lesion to seed throughout the peritoneal cavity [82]. When such shedding occurs, tumor cells initially lose their original ECM and/or cell-to-cell attachments, creating an elevated risk for anoikis and subsequent apoptosis [83]. As a protective mechanism, disseminated OVCA cells cluster in the peritoneal fluid where they form heterotypic aggregates or spheroid-like structures composed of tumor cells and modified stromal cells including CAFs, TAMs, adipocytes and mesothelial cells [84,85]. These multicellular aggregates provide a unique opportunity to assess the contribution of the component cell types to the acquisition of a progressively malignant phenotype.

The isolation of tumor spheroids from OVCA patients with metastatic ascites led to the identification of two subgroups, highly heterotypic and multicellular spheroids, which have a greater metastatic potential than those with no or low numbers of incorporated stromal cells [86]. The various elements within these heterotypic spheroids synthesize ECM proteins that provide the tumor cells with an initial primitive fibronectin–collagen-rich matrix and, thus, additional adhesion-based prosurvival signals. Indeed, upregulation of >700 genes was evident in heterotypic spheroids compared to <30 genes in spheroids formed with OVCA cells alone [86]. The most upregulated include LRP1 (a signaling receptor for PAI-1) [87], uPAR (uPA receptor and a signaling mediator for PAI-1) [88] and PAI-1 itself. These structured niches are more than just havens that facilitate tumor viability. There is growing evidence that heterotypic spheroids in OVCA ascites are highly malignant and increase cancer cell proliferation as well as the overall metastatic burden and are an important conduit for the development of chemoresistance [89,90,91,92]. These heterocellular aggregate structures are designated “metastatic units” because of their unique contributions to tumor spread within the peritoneal cavity and construction of the metastatic niche [86]. Indeed, recent transcriptomic, epigenomic, mass spectroscopy, spatial transcriptomic and multiplexed imaging approaches, at both the bulk tumor and single-cell levels, clarified the complexity of the cellular interactions in the ovarian tumor microenvironment underlying development of resistance to chemotherapy (e.g., [93,94,95]).

While heterotypic intraperitoneal spheroids are drivers of transcoelomic metastasis in high-grade aggressive ovarian cancers [96,97], more current studies identified the critical cellular component in this process as a subpopulation of CAFs (CAFs^high^^α5^) that express elevated levels of integrin α5 (ITGA5). ITGA5 recognizes a corresponding β1 integrin subunit (ITGB1), also upregulated in spheroids from patient ascites [98], to form the functional α5β1 heterodimer. CAFs^high^^α5^ promote tumor cell interaction with the provisional fibronectin–collagen matrix, and α5β1 is involved in spheroid maintenance [86] since targeting heterotypic spheroids, constructed with OVCA cell lines and mesothelial cells, with ITGB1-specific antibodies or ITGB1 siRNA resulted in a loss of spheroid integrity in vitro [98]. Crosstalk between OVCA cells and CAFs^highα5^, moreover, leads to secretion of EGF, which initiates a bidirectional paracrine loop and subsequent increase in tumor cell ITGA5, further stabilizing tumor–stroma interactions [86,99]. Embedded CAFs release EGF, stimulating EGFR-positive cancer cells to produce TGF-β. TGF-β, in turn, further activates fibroblasts to provide tumor prosurvival signals, including elevated expression of ITGA5, while promoting ECM production [92] and upregulation of EGF to maintain persistence of the signaling loop. The relevance of this complex system is highlighted by the findings that inhibition of fibronectin/ITGA5 binding, or interrupting the EGF/EGFR axis, decreases the number of stable spheroids and attenuates OVCA dissemination/invasion in vitro [86] while pharmacologic blockade of the EGF signaling pathway attenuates spheroid formation and OVCA progression in vivo [100].

Cooperation between CAFs^highα5^ and TAMs appears to drive heterotypic spheroid formation and transcoelomic metastasis and both, together with resident cancer stem cells, may contribute to disease recurrence and chemoresistance [101]. Clinical data indicate that elevated expression of ITGA5 correlates with poor outcomes [102], although administration of selective ITGA5 antibodies to patients with later-stage disease failed to provide any protective effect in a phase I pharmacokinetic study [103]. Depletion of CAFs using a PDGFR inhibitor (imatinib) and targeting TAMs (with liposome clodronate), however, disrupted heterotypic spheroids and reduced the overall peritoneal tumor burden in OVCA-implanted mice [86]. This strategy may have clinical significance in solid cancers as well since elimination of CAFs in lung tumors [104] and PDAC [105] decreased metastatic burden; although for PDAC, this is controversial since, in one report, CAF ablation led to the development of invasive, undifferentiated tumors [106].

## 4. CAFs Confer Resistance to Chemotherapy

Interactions between tumors and CAF subpopulations occur at several levels and may have therapeutic implications [14]. Malignant cells direct resident CAFs to express a fibrotic, protumorigenic and niche-forming genetic program. These newly acquired ECM remodeling capabilities likely contribute to the initiation and maintenance of cancer stemness and, in turn, the development of resistance to conventional therapies [68]. Indeed, CAFs play an active and integral role in the development of a drug-resistant phenotype [10,17,107,108]. Coculture of PDAC spheroids with pancreatic stellate cells, for example, increases tumor invasion, ECM remodeling, invadopodia formation, cell plasticity and drug resistance [109]. The involved mechanisms, however, are varied and appear to be both shared as well as exhibit some tumor-type specificity. In highly fibrotic tumors, the intense desmoplastic response, driven in large part by CAFs and TAMs, creates a dense ECM barrier that limits effective drug delivery. The development of a progressively noncompliant fibrotic stroma reduces the chemotherapeutic efficiency of epirubicin, cyclophosphamide and doxorubicin in breast cancer patients [110,111]. Stratification of ovarian malignancies into two subgroups, fibrotic and nonfibrotic, by transcriptome-based approaches, moreover, confirmed that patients with fibrotic signature tumors exhibited significantly shorter overall survival times [112]. This appears to be due, in large part, to the activation of pathways that regulate tumor cell viability and aggressive behavior. Indeed, stromal nicotinamide N-methyltransferase (NNMT) is necessary and sufficient for CAF function (i.e., expression of CAF biomarkers, cytokine secretion, deposition of an oncogenic ECM [113]) in high-grade serous carcinoma [114]. Elevated NNMT expression defined, in part, the metastatic stromal proteome and supports ovarian cancer proliferation, migration and distal spread. CAFs also protect pancreatic cancer cells from gemcitabine-induced apoptosis through a NF-κB→IL-1β→IL-1R/associated kinase-4 (IRAK4) pathway; inhibition of IL-1β, or knockdown of IRAK4, dramatically augment gemcitabine chemosensitivity [115]. IL-6 secretion by esophageal squamous cell carcinoma (ESCC) CAFs, furthermore, increases upon coculture with tumor cells, activating STAT3/NF-κB signaling leading to an induction in CXCR7 expression enhancing chemoresistance to cisplatin [116]. Such drug-induced CAF modulation may be more common than currently appreciated. Gastric cancer cells are protected from chemotherapy-induced cell apoptosis through a CAF-activated IL-6→JAK/STAT3 pathway [117] but only when CAFs are cocultured with gastric tumor cells previously exposed to chemotherapeutics suggesting a drug-dependent mechanism of therapy resistance [117]. Incubation of bladder cancer cells with cisplatin, moreover, increases the transition rate of normal fibroblasts into chronically activated CAFs in the TME. Finally, cisplatin increases expression of PAI-1 by ESCC CAFs leading to an increase in tumor growth and development of chemoresistance [118]. It appears that some drugs commonly used in cancer treatment can have the untoward consequence of engaging new pathways in CAFs that contribute to tumor cell survival, self-renewal and chemoresistance.

It has become apparent, moreover, that multiple CAF-derived cytokines give rise to drug insensitivity perhaps as a function of tumor type. Early studies indicated that conditioned medium (CM) from cultured CAFs imposes enhanced chemoresistance in various malignancies including non-small cell lung cancer (NSCLC) [119,120,121]. CM from cultured CAFs also protect OVCA cells from cisplatin-induced apoptosis [122]; CAF-CM activates STAT3 signaling while increasing expression of the anti-apoptotic protein Bcl-2, a downstream target of STAT3 [123]. The clinical relevance of these findings is highlighted by the fact that inhibition of STAT3 phosphorylation reduces Bcl-2 expression and eliminates the protective effect of CAF-CM on OVCA cells to cisplatin-induced apoptosis [122]. There is abundant evidence implicating individual cytokines (IGFs, interleukins, TGF-β) in acquired chemoresistance. IGF-1 and IGF-2 synthesis by PDAC CAFs confers reduced responsiveness to gemcitabine and paclitaxel in murine models. Treatment of mice with li-gand-blocking IGF antibodies reduced the pool of activated IGF receptors and, in combination with gemcitabine, significantly increased caspase-3 cleavage and the rate of tumor cell death [124]. Gastric cancer CAFs mobilize the PI3K/AKT signaling pathway by production of IL-8, resulting in activation of NF-κB and insensitivity to cisplatin [125]. IL-6 and IL-8 secretion by CAFs in breast cancer appears causative in the generation of resistance to chemotherapy as knockdown of both cytokines re-established drug sensitivity to paclitaxel and doxorubicin [126]. Paracrine interaction between bladder cancer cells and CAFs stimulates synthesis of IGF-1 by tumor cells, upregulating ERβ via an activated IGF-1/IGF-R/AKT/c-Jun signaling axis resulting in an increase in Bcl-2. Silencing ERβ or blocking IGF-1 activity reversed Bcl-2 expression and significantly decreased CAF-promoted resistance to cisplatin in vitro and in vivo [127]. Secretion of IL-6, CXCL1 and the prostaglandin-regulating enzyme COX-2 by stromal CAFs promotes tumor growth in breast and ovarian cancer [128]. Expression of TGF-β by ESCC CAFs, furthermore, confers resistance to multiple chemotherapeutics (cisplatin, carboplatin, docetaxel) [129]. While the various functions of TGF-β in the TME were discussed previously (e.g., [20,130]), analysis of the Protein–Protein Interaction (PPI) network implicated CTGF (CNN2), IGF-1, BMP2, MMP13, TGF-β3, MMP3 and SERPINE1 (PAI-1) as major hub genes in the TGF-β-induced differentiation of human mesenchymal stem cells [131]. The protumorigenic actions of TGF-β likely involve the coordination of multiple signaling pathways in both CAF and non-CAF target cells within the TME. TGF-β signaling engages the EGFR, PDGFR, ERK and AKT/STAT pathways to stimulate cell migration (e.g., Figure 2) as well as activate alternate tumor survival pathways (e.g., elevated expression of the ABC multidrug transporters), resulting in suppression of apoptosis [130].

## 5. CAF-Derived PAI-1 as a Poor Prognosis Biomarker

The clade E member 1 serine protease inhibitor SERPINE1, also known as plasminogen activator inhibitor-1 (PAI-1), is a potent negative regulator of the pericellular proteolytic cascade [3,136,137]. High tumor PAI-1 levels (z-score >2) are constantly prognostic for poor disease outcomes and shorter disease-free survival in various malignancies [138,139,140,141], such as node-negative breast cancer, ovarian serous carcinoma, glioblastoma, renal clear cell carcinoma and gastric cancer as well as head and neck squamous cell carcinoma (HNSCC) [142,143,144,145,146]. PAI-1 expression in the TME is regulated by growth factors, cytokines and hormones including EGF [147,148,149] and TGF-β [150,151,152,153,154,155] via different, perhaps tumor-specific, pathways. In glioblastomas, EGF signals through c-SRC/PKC-δ/sphingosine kinase-1 [156], whereas in breast cancer and OVCA EGF signaling is mediated via NF-κB and ELK1 [157,158]. TGF-β1 also stimulates interaction between phosphorylated receptor SMADs and p53, resulting in the formation of SMAD/p53 transcriptional complexes to activate TGF-β1 target genes [159,160,161], but may also synergize with glucocorticoids (dexamethasone) and the p38MAPK/ERK1/2/SMAD2/3 pathways in OVCA [162]. EGF and TGF-β1 function cooperatively, moreover, to create an aggressive phenotype with upregulation of PAI-1 expression in human cutaneous squamous cell carcinoma [163,164].

SERPINE1 is a member of the validated five-member EMT- or plasticity-related prognostic gene set in gastric cancer and the six-gene signature that accurately predicts reduced overall survival in patients with HNSCC, where it partitions to the aggressive tumor signature subset, indicative of poor prognosis and higher risk score. SERPINE1 is generally classified as a hub or core gene in a wide spectrum of cancer types [50,118,141,146,165,166,167,168,169,170,171,172,173,174,175,176,177,178,179,180,181,182,183]. For many of the poor outcome cancers, identification of SERPINE1 in the complement of hub or signature genes is a strong indicator of reduced patient survival [141,184,185,186,187,188,189]. Furthermore, hypoxia is a characteristic of many solid tumors and, in several cancers, a hypoxic TME is associated with patient mortality [115,190]. SERPINE1 is highly upregulated in such tumors, likely via recognition of hypoxia response elements in the SERPINE1 promoter by HIF-1α and HIF-2α, where it serves as a prognostic hypoxia-associated hub gene [190]. Cytoscape profiling, moreover, similarly implicates SERPINE1 as a major core gene in the genomic program of tissue fibrosis, where it modulates focalized uPA/uPAR-dependent pericellular proteolysis and functions as a signaling activator for LRP1. String Protein– Protein Interaction Network and Gene Ontology analyses confirmed the cooperative role of SERPINE1 and TGF-β1 in the global process of normal and maladaptive (fibrotic) repair; both significantly contribute to the desmoplastic reaction and subsequent enhanced tissue stiffness [32]. This stromal barrier, and the associated increase in interstitial fluid, attenuates the effective delivery of chemotherapeutics to the TME (e.g., [7,191]).

While many cell types in the TME produce PAI-1, CAFs are a prominent source of PAI-1 expression in esophageal malignancies [168]. There is also a strong positive correlation between CAF α-SMA and PAI-1 colocalization in human lung adenocarcinoma [192], and PAI-1 distributes to α-SMA-positive fibroblastoid cells at the invasive margins and stroma-enriched regions in cutaneous squamous cell carcinomas (Figure 3). Since the transition of fibroblasts into CAFs can take different routes, resulting in a diverse group of CAF subtypes, the myofibroblastoid phenotype (α-SMA^high^/PAI-1^high^) is likely involved in the development of chemotherapeutic resistance in cancer cells [162] and/or acquisition of a more aggressive, invasive phenotype [164]. The role of PAI-1 in this process may be more pivotal than previously appreciated. PAI-1 is, in fact, a prominent member of the activated fibroblast gene signature in bleomycin-induced pulmonary fibrosis as well as in serum-stimulated human fibroblasts, and the small-molecule PAI-1 inhibitor SK-216 mitigates TGF-β1-induced myofibroblast differentiation and lung fibrosis [41,193,194]. Similarly, SK-216 dose-dependently attenuates TGF-β-induced expression of α-SMA in both the MLF and MRC-5 lines of human fibroblasts, as well as in CAFs isolated from pleural effusions from lung cancer patients, while reducing CAF viability [192,195,196]. siRNA-mediated PAI-1 knockdown generated a similar result. Collectively, these data suggest that PAI-1 may be a downstream effector of TGF-β1-induced myofibroblast differentiation and the profibrotic genetic program [194]. Notably, proliferation of lung cancer cells cocultured with CAFs significantly increased compared to tumor cells cultured without CAFs; moreover, the apoptotic effect of cisplatin on lung cancer cells cocultured with CAFs is markedly attenuated compared to cells cultured without CAFs [192]. In contrast, incubation of CAFs with SK-216 diminishes CAF α-SMA expression and restores the apoptotic response of CAFs and lung tumor cells to cisplatin. These findings suggest that pharmacologic inhibition of PAI-1 limits drug resistance, perhaps by suppressing the myofibroblastic characteristics of CAFs, and that the PAI-1 functional blockade may be one approach to enhance the efficacy of cisplatin-based chemotherapy in lung cancer cells [192].

## 6. The CAF/PAI-1 Axis in the Tumor Immune Response

Consistent with the growing appreciation for the complex roles of PAI-1 in cancer biology, which transcend the classic roles of control of thrombosis and fibrinolysis, this SERPIN also regulates the immune response in several malignancies. Immunosuppression in the TME is crucial for accelerated tumor growth and disease progression and, in this regard, PAI-1 effectively modulates the immune environment in NSCLC by promoting expression of TGF-β through an IL-6-dependent pathway, as well as the TAM-associated chemo-/cytokines CCL-17, CCL-22 and IL-6 [197]. In an immunosuppressive feed-forward loop, PAI-1 activates the NF-κB→IL-6→STAT3 pathway in TAMs, leading to enhanced TGF-β signaling and subsequently increasing PAI-1 secretion. PAI-1 decreases the number of tumor-infiltrating lymphocytes while stimulating PD-L1 endocytosis in melanoma cells, negatively impacting, thereby, the effectiveness of anti-PD1 antibody immune therapy [198,199].

Analysis of the interplay between hepatocellular carcinoma cells, hepatic stellate cells, CAFs and immune cells established a more complex role for PAI-1 in tumor progression. In this context, liver tumor cells escalate the transition of stellate cells into CAFs and both CAFs and hepatoma cells polarize local TAMs to M2-type macrophages [200]. It appears, moreover, that PAI-1 promotes recruitment and M2 polarization of monocytes/macrophages via different functional domains in the PAI-1 molecule with the LRP1 interaction region facilitating macrophage migration and the uPA recognition domain involved in M2 polarization through a p38→NF-kB→IL-6→STAT3 pathway [201]. Tumor-derived PAI-1 expression is clearly associated, thereby, with increased tumorigenicity, M2 macrophage density and elevated STAT3 signaling, suggesting one possible mechanism for the protumorigenic role for this SERPIN. Moreover, while both CAF- and carcinoma-induced macrophage populations increase the proliferative and metastatic capabilities of hepatocellular carcinoma cells, only the CAF-induced TAMs significantly upregulate PAI-1 expression, which directly correlates with elevated CXCL12 secretion by CAFs; these responses were abolished by CXCL12-specific inhibitory antibodies. Although CXCL12 stimulates PAI-1 transcript expression through the CXCR4 receptor in astroglioma [202], PAI-1 upregulates CXCR4 in CAF-induced TAMs while increasing malignant traits in hepatic tumor cells. PAI-1 functional disruption with the small-molecule inhibitor Tiplaxtinin [203] attenuates the TAM-promoting effects on proliferation, migration and invasion in CAF-induced, but not liver-tumor-cell-induced, TAMs.

## 7. Role of PAI-1 and the CAF/PAI-1 Axis in Tumor Survival and Cell Migration

The PI3K/AKT signaling pathway is a key effector of tumor progression [204,205,206,207,208] and a promising therapeutic target for anticancer therapy (as reviewed in [209]). Al-though the underlying mechanisms may be cell-type-dependent, CAF-associated PAI-1 activates AKT and ERK1/2 via LRP1 signaling stimulating, thereby, squamous cell car-cinoma and macrophage migration and invasive behavior [168]. While ERK1/2 are required signaling effectors in the TGF-β1-induced program of cell migration, other pathways (e.g., EGFR) appear to be involved as well (Figure 2). The complexities of signal integration and the placement of AKT relative to the engagement of the EGFR and MEK/ERK pathways in TGF-β1-dependent cell motility remain to be determined. Expression of PAI-1, moreover, is both induced and regulated by the PI3K/AKT network [210,211], and increased PAI-1 levels activate the PI3K/AKT survival pathway [212]. Exposure of PAI-1-expressing wild-type (WT) or PAI-1-deficient murine fibrosarcoma cells to etoposide confirmed that WT cells are more resistant to drug-induced apoptosis compared to the more sensitive PAI-1-deficient cells, which exhibit a significant downregulation in PI3K/AKT activity. Incubation of PAI-1-expressing WT fibrosarcoma cells with inhibitors to PI3K (Ly294002) or AKT (Akt inhibitor VIII) restores sensitivity to etoposide-induced cell death [212]. Further highlighting the role of PAI-1 in this process, introduction of a PAI-1 expression construct into PAI-1-gene-deficient cells increases both AKT activity and protection against drug treatment, whereas siRNA knockdown of PAI-1 in WT fibrosarcoma cells reduces AKT activity and induces sensitivity to etoposide [212]. These findings are consistent with the significant increase in pAKT levels in cells genetically engineered to overexpress PAI-1 and the induction of both AKT and ERK1/2 phosphorylation in response to exogenous delivery of recombinant PAI-1 protein to serum-starved cells (Figure 4). Recently, a novel PAI-1 inhibitor, ACT001, currently in phase I clinical trials for glioblastoma treatment, attenuated phosphorylation of PI3K and its downstream target AKT, inhibiting U118MG proliferation, migration, invasion and metastasis while triggering a pro-apoptotic response [213,214]. Importantly, similar results accompanied siRNA-induced PAI-1 knockdown. Furthermore, ACT001, when applied together with cisplatin, synergistically mitigated U118MG cell migration and invasion as well as PI3K/AKT phosphorylation, significantly reducing, via enhanced apoptosis, tumor weight and size in an in vivo xenograft model [214]. Elevated expression of PAI-1, therefore, may support tumor growth and cisplatin resistance in glioma cells through the PI3K/AKT pathway and, perhaps more importantly, it appears that PAI-1 is a druggable target, at least in glioblastoma.

While the mechanisms are unclear, the available data suggest that the anti-apoptotic action of PAI-1 plays a crucial role in drug resistance in multiple cancers, likely involving activation of the PI3K/AKT or ERK1/2 signaling pathways, suppression of plasmin generation or caspase-3 activity or regulation of vitronectin-mediated cell adhesion [139,143,216,217]. It appears, moreover, that the PAI-1-mediated rescue from spontaneous apoptosis in response to serum deprivation or plasminogen-induced cell detachment (Figure 5) may be dependent, in part, on control of plasmin activation and/or inhibition of Fas/Fas ligand-mediated cell death [218,219]. These findings may have clinical implications. Esophageal squamous cell carcinoma patients with high levels of PAI-1-expressing CAFs (i.e., 2-4.5-fold relative to patients with low-expressing CAFs) have a significantly worse prognosis than the low-expressing cohort [118]. More importantly, treatment of isolated CAFs with cisplatin stimulates extracellular PAI-1 accumulation and engagement of a paracrine signaling loop in which PAI-1 activates the PI3K/AKT and ERK1/2 pathways and inhibits caspase-3. siRNA-induced downregulation of PAI-1 in ES2 human OVCA cells, as well as pharmacological inhibition of ES2-secreted PAI-1 activity by the small-molecule inhibitor of PAI-1 TM5275 [220], results in G2/M cell cycle arrest, an increase in caspase-3 activity and reduced caspase-8 levels, indicating a downregulation of the extrinsic apoptotic pathway. The accompanying increase in cytochrome C release, a biomarker of mitochondrial damage [139], additionally suggests involvement of the intrinsic apoptotic pathway. Furthermore, TM5275 blocks PAI-1 binding to its signaling receptor, LRP1 [221], highlighting a potential role for LRP1-mediated signaling in these PAI-1-dependent processes. In corresponding pairs of paclitaxel-resistant and parental MDA-MB-231 and MCF-7 human breast cancer cell lines [222,223], PAI-1 is greatly upregulated in drug-resistant cells [224]. shRNA knockdown of PAI-1 upregulates cleaved caspase-3, induces apoptosis and attenuates cell survival in vitro. PAI-1 knockdown in paclitaxel-resistant cells also significantly reduces tumor growth in mice, suggesting a critical role for PAI-1 in maintaining paclitaxel resistance in breast cancer [224]. Collectively, these findings underscore the contribution of PAI-1 to the acquisition of the aggressive phenotype and shed light on the consistent inclusion of PAI-1 among the biomarkers of poor prognosis and reduced disease-free survival times in cancer patients. Elevated PAI-1 expression, moreover, stimulates migration of tumor cells as well as macrophages and increases tumor invasion through a LRP1/PI3K/AKT signaling cascade, while incubation of ESCC cells with Tiplaxtinin attenuates PI3K and AKT phosphorylation and reduces resistance to cisplatin in vitro. As a proof of principle, mice were implanted with ESCC cells combined with either a control or PAI-1-expressing NIH3T3 cells prior to xenografting and administration of Tiplaxtinin via oral gavage [118]. Cotreatment with cisplatin and Tiplaxtinin reduced tumor size in vivo, suggesting that targeting PAI-1 in ESCC during cisplatin infusion may be a promising therapeutic approach [118].

Drug-induced changes to CAFs and/or tumor cells, however, may have unanticipated consequences. Incubation of the cisplatin-resistant ovarian cancer cell line A2780cp with carboplatin increases expression and secretion of PAI-1 and is accompanied by a strong EMT response (e.g., reduction in E-cadherin, upregulation of mesenchymal markers vimentin, Snail and Twist, increased cell migration and invasion in vitro) [227]. Conversely, siRNA-mediated knockdown of PAI-1 in A2780cp cells inhibits EMT and increases sensitivity to carboplatin. PAI-1 also plays a critical role during the mesothelial-to-mesenchymal transition (MMT) in OVCA, which involves the transdifferentiation of mesothelial cells into cancer-associated mesothelial (CAM) cells [228]. In vitro experiments with human ovarian tumor cell lines (ES2, SKOV3, Hey), all expressing PAI-1, confirmed that PAI-1 initiates the formation of CAMs by activating the NF-kB signaling pathway in mesothelial cells. CAMs, in turn, express several cytokines, including IL-8 and CXCL5, establishing a positive feedback loop and metastatic phenotype in OVCA cells that is inhibited by shRNA-based knockdown of PAI-1 in CAMs [229]. These findings are clinically relevant as IL-8 and CXCL5 both increase metastasis in OVCA cells [230,231]. Similarly, when untreated and PAI-1 shRNA-treated ES2 OVCA cells were cocultured with human explanted omentum, PAI-1 knockdown significantly mitigates cell invasion into the omental tissue; this inhibition could be reversed by addition of recombinant PAI-1 [229].

PAI-1 is a major downstream TGF-β1 target gene, and both PAI-1 and TGF-β1 promote tumor cell aggressiveness and tissue invasion, epithelial migration and amoeboid motility [165,167] (Figure 6). TGF-β1-induced cell mobility, moreover, is effectively and dose-dependently attenuated by Tiplaxtinin (Figure 6), suggesting that the PAI-1 functionis required for TGF-β1-stimulated locomotion, which is consistent with the finding that PAI-1 or TGF-β1 deficiency reduces cell motility [225]. Expression of a PAI-1-GFP, fusion protein under the inducible control of +800 bp of the injury-activated PAI-1 promoter prominently “marks” keratinocyte migration trails, while the use of PAI-1-null cells, knockdown approaches, PAI-1 add-back rescue and neutralizing antibodies confirmed the requirement for PAI-1 in cell movement [225]. Addition of active recombinant PAI-1 to wounded wild-type keratinocyte monolayers as well as to PAI-1^−/−^ MEFs and PAI-1^−/−^ keratinocytes significantly stimulates directional motility above basal levels in all cell types, while the attenuated migratory activity as a consequence of antisense-mediated PAI-1 downregulation is effectively reversed by addition of recombinant PAI-1. Since CAFs are a major source of PAI-1 in the immediate TME of several aggressive malignancies [168,192], it may well be that PAI-1 signals in a paracrine fashion to promote tumor aggressiveness, chemoresistance and tissue invasion [162]. Indeed, addition of CAF conditioned medium stimulates tumor cell migration in response to monolayer wounding compared to monolayers receiving medium from normal fibroblasts (Figure 6). PAI-1 functional blockade may be one approach to enhance the efficacy of cisplatin-based chemotherapy in lung cancer cells [192].

## 8. Future Directions

CAFs are prominent members of the complement of cellular elements in the TME, coexisting in a dynamic interaction with the growing tumor mass as well as with various resident and recruited cell types. CAFs facilitate tumor initiation and progression, foster cancer cell plasticity and stemness, stimulate stromal remodeling and contribute to the acquisition of the highly lethal drug-resistant metastatic phenotype. The considerable heterogeneity in CAF subsets likely reflects such differential functions and is underscored by the wide spectrum of biomarkers that characterize protumorigenic CAFs (Table 1; adapted from [233]).

The construction of an increasingly fibrotic stroma, a characteristic of desmoplastic tumors largely under the direction of CAF-produced growth factors and cytokines, serves as a barrier to limit accessibility of chemotherapeutics to the TME, enabling tumor cell survival and proliferation as well as the development of aggressive behavior and an attenuated response to therapy (Table 2; adapted from [12,44]). Multiple CAF-derived cytokines contribute to drug resistance. In many cases, these target specific genes causally involved in the development of chemo-insensitivity. SERPINE1 or PAI-1 is one such candidate and a prominent core or hub gene in various cancer types, where the level of expression is a predictor of patient outcomes. PAI-1 regulates pericellular proteolysis and, thereby, TME mechanics and barrier status. This multifunctional protease inhibitor also impacts the tumor immune response [197] through the reprogramming of immune cells newly recruited into the TME, enhances tissue invasive traits and modulates tumor survival pathways (Figure 7). The combination of in vitro and in vivo approaches confirmed the regulatory role of PAI-1 in cellular auto- and paracrine signaling processes within the TME. These findings support the conclusion that PAI-1 is, in fact, a major contributor to the drug-resistant phenotype and that targeting PAI-1 function with small-molecule inhibitors may restore chemo-sensitivity. In this regard, the development of the CAF/tumor cell 3D coculture model [234,235] presents an opportunity to identify additional paracrine acting factors and pathways that contribute mechanistically to cancer progression. This system, designed using specified biomaterials and component cell types, approximates the complexity of the in vivo TME. Such near-physiological tissue constructs may provide a platform to clarify the cooperative and bidirectional signaling processes underlying CAF–tumor cell interaction as well as define new targets for drug-based therapies [236]. The available data suggest that the combined CAF–cancer cell model may have clinical utility both in assessing the efficacy of established chemotherapeutics in a more pathophysiologic context and as a screening tool for drug discovery [234,235,237,238]. 

Such coculture systems may also be adapted to delineate the basis for biochemical crosstalk among tumor and stromal cells. Indeed, it is now evident that CAFs secrete metabolites (e.g., lactate, pyruvate, amino acids, free fatty acids), which are incorporated by their neoplastic neighbors and support tumor progression [239,240,241,242]. Export of metabolites can also occur through exosomal transport [243,244,245]. In response to cancer-derived factors, CAFs rewire their metabolic profile to overproduce and secrete those metabolites, which will be reabsorbed rapidly by cancer cells due to upregulation of required transporters in the tumor cell plasma membrane [246,247,248,249]. In breast cancer cells, stromally expressed TGF-β not only leads to anticipated fibroblast activation and ECM production but can also stimulate autocrine signaling in CAFs, leading to a shift in catabolic metabolism, oxidative stress and increased aerobic glycolysis [250]. CAF metabolic reprogramming also induces epigenetic changes to maintain the CAF activation state and its protumorigenic function. Anabolic cancer cells and catabolic CAFs, therefore, are metabolically coupled. This metabolic dialogue between tumor cells and CAFs is an important contributor to the development of an aggressive chemo-resistant tumor phenotype and reflects the increased need for nutrients and metabolic intermediates to support anabolic processes in the growing tumor [251,252,253,254,255] while modulating the immune response [256,257,258]. Therapeutic approaches aimed at deactivating myofibroblast-type CAFs could be a promising strategy to interrupt the metabolic connection between those CAFs and cancer cells in the TME [259].

Recently, the application of microRNA technology has expanded the repertoire of PAI-1-focused therapeutics. MicroRNAs are short sequences of noncoding RNAs (~20 nucleotides) that affect expression of specific target genes by binding to their 3′ untranslated region (3′-UTR). Single miRNAs can target multiple transcripts, and several may be useful in cancer diagnostics and therapy as well as provide new biomarkers to monitor disease progression [260,261,262,263,264,265]. Conversion of normal fibroblasts into CAFs is accompanied by up- or downregulation of specific miRNAs [57,265,266,267,268] that facilitate communication between “activated” fibroblasts and cancer cells [269]. Since dysregulation of CAF-specific miRNAs affects tumor cell growth, migration, invasion and adaptation to chemotherapy [141,270], pharmaceutical approaches utilizing miRNAs and the miRNA-mediated transition of tissue fibroblasts to CAFs is the focus of several recent clinical trials [264]. Certain miRNAs that target the 3’-UTR of PAI-1 mRNA transcripts, identified using TargetScanHuman, mitigate tumor growth and progression in breast, bladder and gastric cancers and osteosarcoma [142]. Progressive hypoxia in the fibrosing TME also increases PAI-1 expression via reductions in miRNA 449a/b [270,271]. Similarly, in bladder cancer, the miRNA-143/-145 cluster, which recognizes the 3’-UTR of PAI-1, is attenuated in all disease stages leading to PAI-1 upregulation [272], while in human osteosarcoma mouse models injection of miRNA-143 suppressed lung metastasis [273]. PAI-1 downregulation by targeting siRNA results in lower expression and secretion of MMP-13 and elimination of lung metastasis [274]. These findings are consistent with the realization that the most upregulated plasticity-associated gene expressed in the TME, which confers a highly aggressive, invasive phenotype, encodes PAI-1 [274,275].

## 9. Conclusions

Various CAF subpopulations support tumor progression and cancer cell phenotypic transitions largely through paracrine signaling by a diverse complement of secreted growth factors and cytokines. This cooperative dialogue between CAFs and cancer stem cells promotes cellular plasticity and tumor progression, maintenance of cancer cell stemness, metabolic reprogramming, extracellular matrix (ECM) remodeling and metastasis. Clarifying the critical pathways in the CAF–tumor interacting network may provide new venues for therapeutic intervention. The mounting evidence of CAF heterogeneity [233] and protumorigenic functional complexity, including their ability to continuously modify the stromal structure of the TME through several cooperative SMAD/non-SMAD pathways, suggests that the development of targeted therapies will require a multidimensional approach. Exploring new strategies for the therapeutic use of specific miRNAs in the TME, perhaps in combination with pharmacologic approaches to inhibit the function of key tumor progression genes (e.g., SERPINE1), may have therapeutic utility and improve patient outcomes [132,269,276].

## Figures and Tables

**Figure 1 cancers-14-01231-f001:**
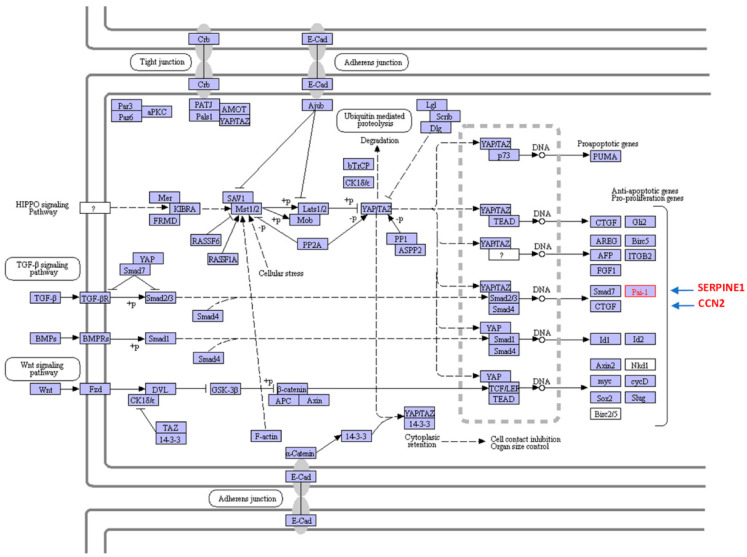
KEGG Pathway map illustrating crosstalk between the TGF-β1 and the Hippo pathways leading to expression of the profibrotic SERPINE1 (PAI-1) and CCN2 (CTGF) genes. TGF-β1 activates a canonical signaling network involving SMAD2/3-dependent transcription of SERPINE1 and CCN2. Noncanonical pathway engagement in response to TGF-β1 (e.g., Hippo) stimulates YAP/TAZ activation and nuclear translocation that cooperate with the TGF-βR-phosphorylated SMAD2/3 transcriptional effectors and the shuttle SMAD4 to induce high-level SERPINE1 (PAI-1) and CCN2 (CTGF) expression. Adapted from the KEGG integrated database [36].

**Figure 2 cancers-14-01231-f002:**
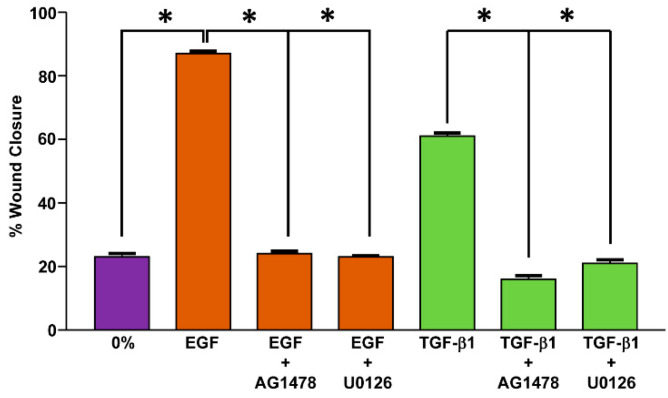
TGF-β1 requires EGFR and ERK signaling to promote a migratory phenotype. Confluent monolayers of T2 epithelial cells were scrape-injured prior to addition of serum-free medium (0%) or serum-free medium containing EGF (10 ng/mL) or TGF-β1 (1 ng/mL) with or without the EGFR kinase inhibitor AG1478 or the MEK1/2 inhibitor U0126. While ERK1/2 are required signaling effectors in TGF-β1-stimulated cell migration [132], TGF-β1 also transactivates the EGFR, likely upstream of MEK/ERK engagement, as part of the cellular motile program [133,134]. Pharmacological blockade with AG1478 or U0126 effectively mitigated both EGF- and TGF-β1-stimulated motility. These same pathways are involved in TGF-β1-induced expression of the promigratory SERPIN PAI-1. Modified from [132,133,134,135]. Data plotted are the mean ± SD of 3 independent experiments. Asterisks = *p* < 0.001.

**Figure 3 cancers-14-01231-f003:**
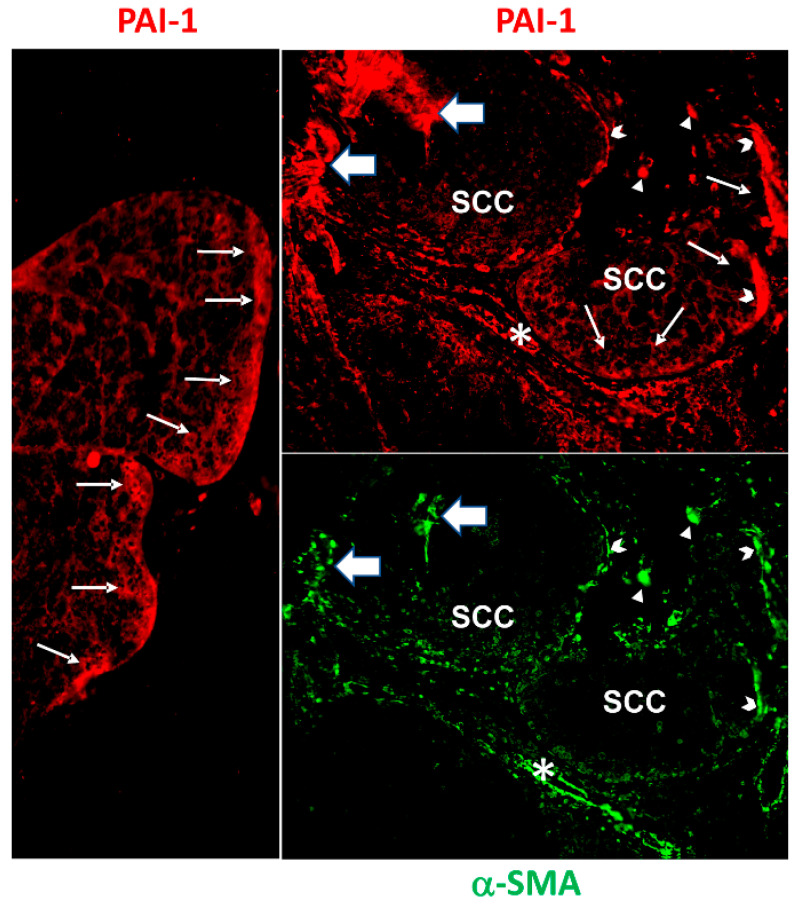
Immunohistochemical localization of PAI-1 (**red**) and α-smooth muscle cell actin (**green**) in early invasive cutaneous squamous cell carcinoma (SCC). PAI-1-positive cells are evident at the tumor invasive margins (thin arrows in left and top-right panels), in large fibroblastoid elements (barbed arrowheads, top-right panel) and in complex stromal–cellular PAI-1 aggregates (large arrows, top-right panel). Examples of significant colocalization of PAI-1- and α-smooth muscle cell actin (α-SMA)-positive fibroblastoid cells (barbed arrowheads, top- and bottom-right panels, respectively) is depicted. Barbed arrows indicate PAI-1/α-SMA expression in cells at the tumor periphery, while arrow heads depict PAI-1/α-SMA in the stroma. Asterisks identify vascular structures (in top- and bottom-right panels). Modified from [163].

**Figure 4 cancers-14-01231-f004:**
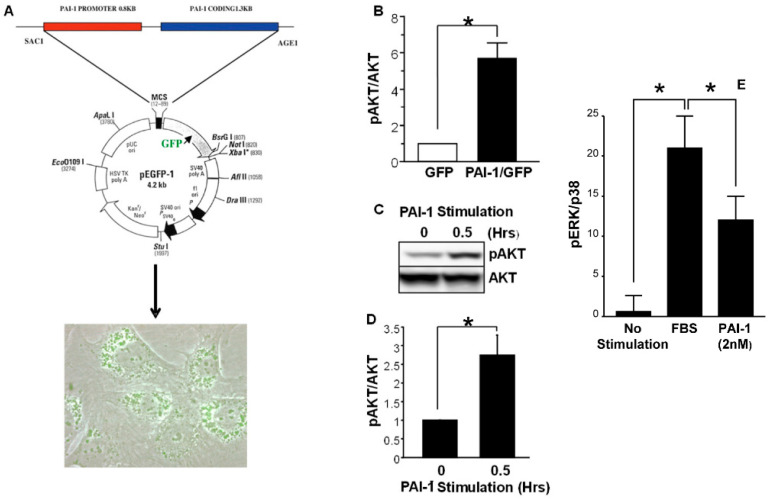
Vector-driven expression and exogenous addition of recombinant PAI-1 protein stimulates AKT and ERK1/2 phosphorylation. A pEGFP-1-based expression construct, in which a chimeric transcript consisting of 1.3 kb of human PAI-1 coding sequences and GFP is transcribed under the control of a 0.8 kb PAI-1 promoter segment, was transfected into R22 cells and stable expressing clones selected (**A**). pAKT levels in chimera transfectants were 6- to 8-fold those of GFP-only vector controls (**B**). Addition of recombinant PAI-1 (20–40 nM) to serum-starved cells induces a rapid (within 30 min) and transient increase in AKT (**C**,**D**) and ERK1/2 (**E**) phosphorylation. Asterisks = *p* < 0.001; Modified from [215].

**Figure 5 cancers-14-01231-f005:**
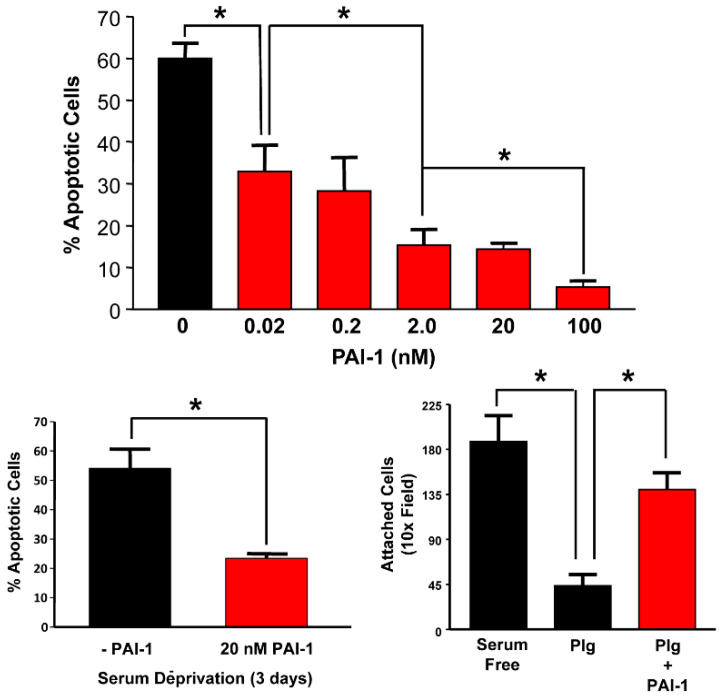
PAI-1 rescues epithelial cells from serum deprivation-induced apoptosis and plasminogen-mediated substrate detachment. T2 cells undergo a significant apoptotic response (involving >60% of the population) following a 3-day incubation in serum-free medium. Addition of the 14-1b recombinant PAI-1 protein (but not BSA), in final concentrations of 0.02 to 100 nM at the time of change-over to FBS-deficient medium, dose-dependently protected T2 cells from apoptosis due to the stress of serum removal (**top panel**). The significant (*p* < 0.005) PAI-1-induced apoptotic “rescue” evident in cultures incubated in 20 nM PAI-1 (**bottom-left panel**) correlated with the PAI-1- related increase in AKT/ERK1/2 phosphorylation (Figure 4) and is consistent with previous observations that PAI-1 initiates signaling events in various cell types. The rapid plasminogen (Plg)-induced detachment of human cutaneous squamous carcinoma cells from the culture substratum and loss of viability was significantly attenuated by simultaneous addition of recombinant PAI-1 (**bottom, right panel**). The survival index for Plg+PAI-1 cultures was >4-fold that of non-PAI-1-treated keratinocytes, suggesting that exogenous PAI-1 protected cells from Plg-induced anoikis. Data plotted are the mean ± SD of triplicate independent experiments. Asterisks = *p* < 0.05. Modified from [225,226].

**Figure 6 cancers-14-01231-f006:**
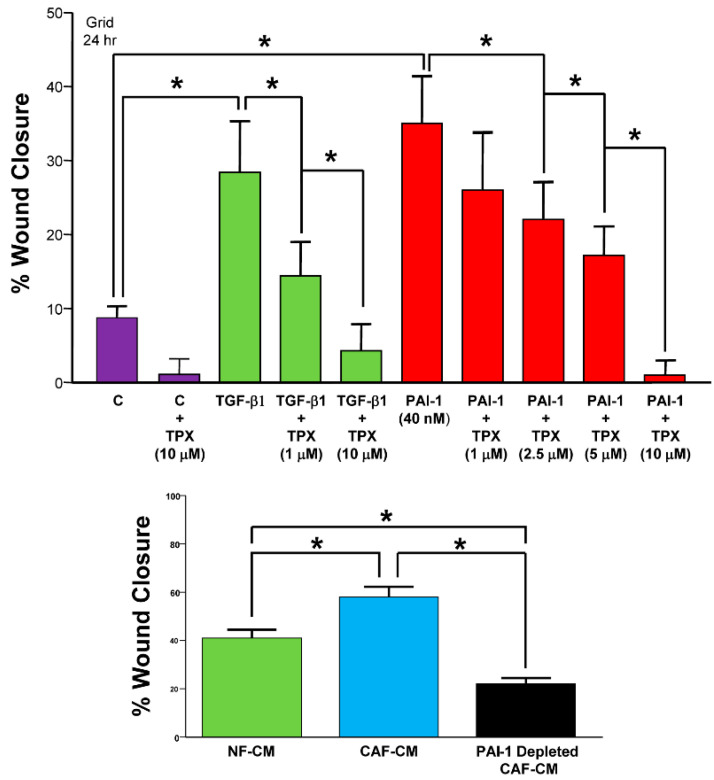
Tiplaxtinin inhibits TGF-β1- and PAI-1-stimulated cell migration in response to monolayer scratch wounding. Confluent monolayers of R22 cells were incubated with the indicated concentrations of Tiplaxtinin (TPX) for 30 min prior to zonal denudation and addition of TGF-β1 or recombinant PAI-1. Injury site closure was measured 24 h later using a calibrated ocular grid (top panel). Data plotted are the mean ± SD of multiple independent assessments. Moreover, addition of CAF serum-free conditioned medium (CAF-CM), derived from cultures of fibroblastoid cells isolated from tumors arising upon implantation of malignant mouse keratinocytes, stimulated motility of immortalized RK rat keratinocytes in response to monolayer wounding compared to cells receiving medium from normal dermal mouse fibroblasts (NF-CM) (bottom panel). Immunodepletion of PAI-1 from CAF-CM significantly reduced cell locomotion. These data are consistent with the PAI-1 requirement for optimal planar migration in RK cells [225]. Asterisks = *p* < 0.05. Modified from [215,232].

**Figure 7 cancers-14-01231-f007:**
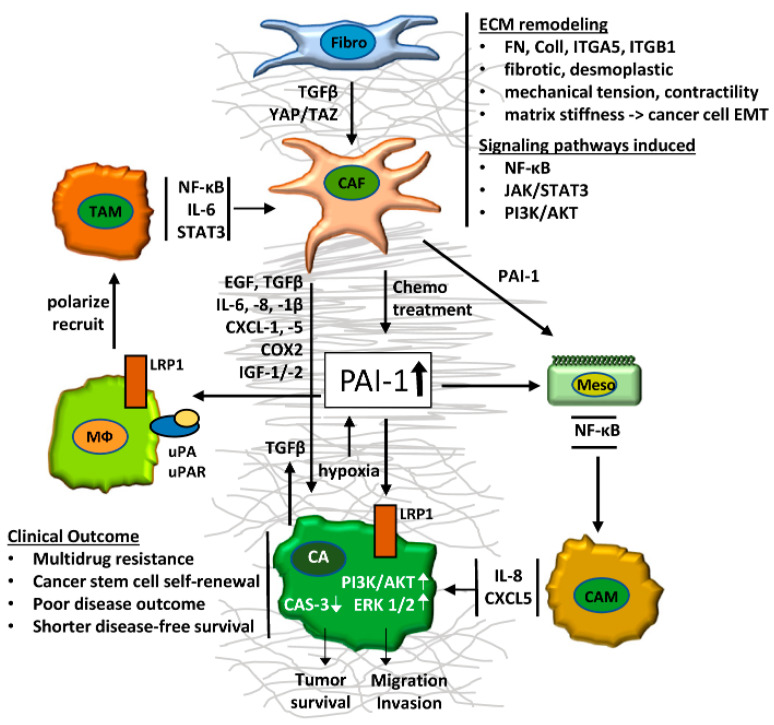
CAF-derived PAI-1 is a central regulator of multicellular functions in the TME. Modulation of the TME includes paracrine signaling circuits between cancer-associated fibroblasts (CAF) and cancer cells (CA) through GF-driven (TGF-β, EGF, IGF-1, IGF-2) and cytokine- (IL-1β, IL-6, IL-8) and chemokine-stimulated (CXCL-1, CXCL-5) expression and secretion of PAI-1. CAF-released EGF stimulates EGFR-dependent TGF-β secretion in malignant cells through a paracrine signaling loop, additionally transdifferentiating resident fibroblasts to CAFs, thereby ultimately further increasing extracellular PAI-1 levels. PAI-1 promotes the recruitment and polarization of M2-type macrophages (MΦ) to TAMs and the transition of mesothelial cells (Meso) to cancer-associated mesothelial (CAM) cells. Infiltration of the TME by modified stromal cells initiates remodeling of the tumor ECM through increased secretion of fibronectin (FN) and collagen (Coll) and expression of specific ECM receptors (ITGA5, ITGB1), ultimately generating a modified fibrotic desmoplastic TME. Increased mechanical tension and CAF-asserted contractility of this tumor matrix induces signaling pathways (e.g., NF-κB, JAK/STAT3), in conjunction with the direct interaction of PAI-1 with cell surface receptors (LRP1, uPAR) on inflammatory and cancer cells. Collectively, these events stimulate survival pathways (PI3K/AKT, ERK1/2) and block pro-apoptotic signals (caspase-3), leading to multidrug resistance, cancer stem cell self-renewal and metastatic spread. Chemotherapy itself can prompt production of PAI-1 by CAFs and thereby activate mechanisms protecting neighboring cancer cells from such drugs. Ultimately, activated expression and increased accumulation of extracellular PAI-1 in the multicellular, fibrotic TME will manifest a poor disease outcome and be prognostic for shorter disease-free survival in patients with epithelial malignancies. Fibro = fibroblast; CAF = cancer-associated fibroblast; CA = cancer cell; Meso = mesothelial cell; CAM = cancer-associated mesothelial cell; MΦ = macrophage; TAM = tumor-associated macrophage; FN = fibronectin; Coll = collagen).

**Table 1 cancers-14-01231-t001:** CAF biomarkers and protumorigenic functions.

Marker	Function
FAP	ECM remodeling, protease activity
PDGFα,β	Receptor tyrosine kinase, angiogenesis, immune cell modulation
CD10	Protease activity, cancer stemness, drug resistance
CD74	Protein trafficking, chaperone activity, immunomodulation
GRP77	Proinflammatory signaling, tumor stemness
α-SMA	Cell contractility, proliferation, desmoplasia
S100A4	Tumor cell migration, proliferation, fibrosis, metastasis
PDPN	Cell motility
CD70	T-cell regulation, tumor cell migration
Vimentin	Cytoskeletal organization, cell motility, tumor invasion
POSTN	ECM remodeling
CDK1	Cell cycling

**Table 2 cancers-14-01231-t002:** Bioactive cytokines and growth factors secreted by CAFs.

Mitogens	Proinvasion Factors
EGF	HGF
HGF	TGF-β
IGF-1	CCL5
SDF-1	POSTN
FGF-1	IL-6
FGF-3	IL-11
**Inflammatory Mediators**	**Stemness Factors**
IL-1	IL-6
IL-6	IL-8
IL-8	HGF
IL-11	IGF-2
LIF	POSTN
Various chemokines	
**Proangiogenic Factors**	**Survival Factors**
VEGFA	IGF-1
SDF-1	IGF-2
FGF-2	IL-6
IL-8	OPN
PDGF-C

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
