# Peer review of "Cancer-Associated Fibroblasts: Mechanisms of Tumor Progression and Novel Therapeutic Targets"

_cancers, 2022, doi:10.3390/cancers14051231_

Round 1

Reviewer 1 Report

This review is an overview of cancer-associated fibroblasts (CAF) function or characterization for cancer therapy. The tumor microenvironment has been recently noted and is an essential concept in cancer. Therefore, this review is useful for researchers on cancer biology. However, some important concept is poor or lack in this review. Especially, in vitro model of cancer-CAF for drug screening has been recently reported based on the cancer-CAF interaction. The tool is helpful for the development of CAF-targeting drugs.

Taken together, major revisions should be made before re-submission. This paper would be accepted only when all the comments are responded to.

  1.  

The CAF surface markers or genes should be shown in the new Table.

2.

3D in vitro model should be added in a new section or future perspective. Because the title includes cancer therapy, in vitro model of cancer-CAF is essential for drug screening. The authors should explain the concept by quoting the related recent references. I suggest at least these references be added for readers’ better understanding.

Review article (for concept)

Cancers 202012(10), 2754

Research paper

Acta Biomaterialia 75 (2018) 200–212

Tissue Eng. Part C Methods 201925, 711–720 https://doi.org/10.1089/ten.tec.2019.0189

Acta Biomaterialia 132 (2021) 448–460

  1. The function of cytokines secreted from CAF should be summarized in table.

Author Response

We would like to express our sincere appreciation for the very helpful comments provided by Reviewer #1. You will see that we revised our manuscript in accord with each suggestion. The specific changes made are as follows:

1 Newly included Table 1 (on page 15) now lists the CAF marker proteins as requested and reference 276 is cited as the source of this information.

2. A discussion of the CAF/tumor cell 3D model is now included in the Future Directions section and appears on page 15. We have also included as citations the review article and the 3 research papers suggested by the Reviewer in this discussion. We also concluded (now in the revised text) that "The available data suggest that the combined CAF/cancer cell model may have clinical utility both in assessing the efficacy of established chemotherapeutics in a more pathologic context and as a screening tool for drug discovery [226, 228, 229]."

3.  A new table describing the functions of the cytokines and growth factors secreted by CAFs is now included as requested by the Reviewer on page 16 of our revised manuscript. It is also indicated in the revised text that the source of this information is references [12, 43]..

Again, we thank Reviewer #1 and hope the Editor will find our manuscript appropriately revised.

Reviewer 2 Report

This is a very interesting manuscript. It is a very informative and well-organized short review, focusing on the complex interactions between CAFs and the various populations of normal and neoplastic cells within tumor microenvironment. The topic is clearly of interest and relevant for research on pre-clinical and clinical level.

Although the proteins (including cytokines, chemokines and other regulators), released into the tumor microenvironment by cancer cells, immune cells and CAFs is certainly extremely important, recent works show that also the metabolites released by all the cells, constituting TME, can also act as important modulators of tumor growth or resistance to therapy.

Therefore, I would like the authors to insert a brief paragraph on metabolites released into the tumor microenvironment and on effects these may have on tumor immune response. As reference the authors could use the article PMID: 34604232

Author Response

We would like to express our appreciation to Reviewer #2 for suggesting that we add a paragraph on the role of secreted metabolites. We now include in this revised manuscript a discussion of metabolites released by CAFs and other accessory cells within the tumor microenvironment and their effects on tumor phenotype, chemoresistance and as a modulator of the immune response. We have included not only the reference requested by the reviewer but we cited an additional 15 relevant.references .    

Reviewer 3 Report

The authors have done a commendable job by compiling a comprehensive review titled"Cancer-Associated Fibroblasts:Mechanisms of Tumor Progression and Novel Therapeutic Targets".

The article is very well structured and organized with relevant diagrams to support the literature.

This review will be of importance to researchers studying tumor-microenvironment and related research.

There is insightful addition of PAI-1as a poor prognosis biomarker and the CAF/PAI-1 axis in the tumor immune response.

Overall, it is an updated and  pertinent research review and would request the Editor to accept the manuscript for publication.

Author Response

We certainly appreciate the gracious assessment of our manuscript by Reviewer #3.

Round 2

Reviewer 1 Report

The authors have responded to all comments. Therefore, I recommend the publication.